# USP30 sets a trigger threshold for PINK1–PARKIN amplification of mitochondrial ubiquitylation

Emma V Rusilowicz-Jones[1],* , Jane Jardine[1],* , Andreas Kallinos[1] , Adan Pinto-Fernandez[2], Franziska Guenther[3] , Mariacarmela Giurrandino[3], Francesco G Barone[1] , Katy McCarron[1], Christopher J Burke[4], Alejandro Murad[4], Aitor Martinez[1] , Elena Marcassa[1], Malte Gersch[5,6], Alexandre J Buckmelter[4], Katherine J Kayser-Bricker[4], Frederic Lamoliatte[10], Akshada Gajbhiye[10], Simon Davis[2] , Hannah C Scott[2], Emma Murphy[3], Katherine England[3], Heather Mortiboys[7] , David Komander[8,9], Matthias Trost[10], Benedikt M Kessler[2], Stephanos Ioannidis[4], Michael K Ahlijanian[4], Sylvie Urbé[1] , Michael J Clague[1]

The mitochondrial deubiquitylase USP30 negatively regulates the selective autophagy of damaged mitochondria. We present the characterisation of an N-cyano pyrrolidine compound, FT3967385, with high selectivity for USP30. We demonstrate that ubiquitylation of TOM20, a component of the outer mitochondrial membrane import machinery, represents a robust biomarker for both USP30 loss and inhibition. A proteomics analysis, on a SHSY5Y neuroblastoma cell line model, directly compares the effects of genetic loss of USP30 with chemical inhibition. We have thereby identified a subset of ubiquitylation events consequent to mitochondrial depolarisation that are USP30 sensitive. Within responsive elements of the ubiquitylome, several components of the outer mitochondrial membrane transport (TOM) complex are prominent. Thus, our data support a model whereby USP30 can regulate the availability of ubiquitin at the specific site of mitochondrial PINK1 accumulation following membrane depolarisation. USP30 deubiquitylation of TOM complex components dampens the trigger for the Parkin-dependent amplification of mitochondrial ubiquitylation leading to mitophagy. Accordingly, PINK1 generation of phospho-Ser65 ubiquitin proceeds more rapidly in cells either lacking USP30 or subject to USP30 inhibition.

## Introduction

Damaged mitochondria are removed from the cell by a process of selective autophagy termed mitophagy. Defects in mitochondrial turnover have been linked to a number of neurodegenerative conditions, including Parkinson's disease (PD), Alzheimer's disease, and motor neuron disease (Sorrentino et al, 2017; Fritsch et al, 2019). This process is best understood in the context of PD, for which loss of function mutations in the mitophagy promoting genes *PINK1* and *PRKN* (coding for the Parkin protein) are evident (Pickrell & Youle, 2015; Bingol & Sheng, 2016). Mitochondrial depolarisation leads to the accumulation of the PINK1 kinase at the mitochondrial surface, which then phosphorylates available ubiquitin moieties at Ser65 (Kane et al, 2014; Koyano et al, 2014; Ordureau et al, 2014; Kazlauskaite et al, 2014b; Wauer et al, 2015b). Phospho-Ser65 ubiquitin (pUb) recruits the ubiquitin E3 ligase Parkin to mitochondria, where it is fully activated by direct PINK1-dependent phosphorylation at Ser65 of its ubiquitin-like (UBL) domain (Jin & Youle, 2013; Kazlauskaite et al, 2014a; Wauer et al, 2015a; Gladkova et al, 2018). This triggers a feed-forward mechanism that coats mitochondria with ubiquitin, leading to selective engulfment by autophagosomal membranes (Harper et al, 2018; Pickles et al, 2018).

The deubiquitylase (DUB) family of enzymes plays a role in most ubiquitin-dependent processes, by promoting ubiquitin flux or

---

[1]Department of Cellular and Molecular Physiology, Institute of Translational Medicine, University of Liverpool, Liverpool, UK    [2]Target Discovery Institute, Nuffield Department of Medicine, University of Oxford, Oxford, UK    [3]Alzheimer's Research UK, Oxford Drug Discovery Institute, Target Discovery Institute, Nuffield Department of Medicine, University of Oxford, Oxford, UK    [4]FORMA Therapeutics, Watertown, MA, USA    [5]Chemical Genomics Centre, Max-Planck-Institute of Molecular Physiology, Dortmund, Germany    [6]Department of Chemistry and Chemical Biology, Technische Universität Dortmund, Dortmund, Germany    [7]Sheffield Institute for Translational Neuroscience (SITraN), University of Sheffield, Sheffield, UK    [8]Ubiquitin Signalling Division, Walter and Eliza Hall Institute of Medical Research, Parkville, Australia    [9]Department of Medical Biology, University of Melbourne, Melbourne, Australia    [10]Laboratory for Biological Mass Spectrometry, Newcastle University Biosciences Institute, Faculty of Medical Sciences, University of Newcastle, Newcastle, UK

Correspondence: urbe@liv.ac.uk; clague@liv.ac.uk
Christopher J Burke's present address is Yumanity Therapeutics–Discovery Biology, Cambridge, MA, USA
Alejandro Murad's present address is Skyhawk Therapeutics, Neurobiology, Waltham, MA, USA
Katherine J Kayser-Bricker's present address is Halda Therapeutics, Branford, CT, USA
Alexandre J Buckmelter's present address is Morphic Therapeutic, Waltham, MA, USA
Stephanos Ioannidis's present address is H3 Biomedicine, Cambridge, MA, USA
Michael K Ahlijanian's present address is Pinteon Therapeutics, Newton, MA, USA
*Emma V Rusilowicz-Jones and Jane Jardine contributed equally to this work

---

suppressing ubiquitylation of specific substrates (Clague et al, 2013, 2019). USP30 is one of only two DUBs that possess a transmembrane domain. Its localisation is restricted to the outer mitochondrial membrane (OMM) and to peroxisomes (Nakamura & Hirose, 2008; Urbe et al, 2012; Marcassa et al, 2018; Riccio et al, 2019). USP30 can limit the Parkin-dependent ubiquitylation of selected substrates and depolarisation-induced mitophagy in cell systems that have been engineered to overexpress Parkin (Bingol et al, 2014; Cunningham et al, 2015; Liang et al, 2015; Hoshino et al, 2019). We have recently shown that it can also suppress a PINK1-dependent component of basal mitophagy, even in cells that do not express Parkin (Marcassa et al, 2018). Thus, USP30 may represent an actionable drug target relevant to PD progression and other pathologies to which defective mitophagy can contribute (Bravo-San Pedro et al, 2017; Tsubouchi et al, 2018; Miller & Muqit, 2019). One attractive feature of USP30 as a drug target in this context is that its loss is well tolerated across a wide range of cell lines (Meyers et al, 2017).

The ubiquitin-specific protease (USP) DUB family are cysteine proteases and comprise around 60 members in humans (Clague et al, 2019). Early academic efforts to obtain specific small molecule inhibitors were only partially successful (Ritorto et al, 2014). More recently, industry-led efforts have generated some highly specific inhibitors, exemplified by compounds targeting USP7, an enzyme linked to the p53/MDM2 signalling axis (Kategaya et al, 2017; Lamberto et al, 2017; Turnbull et al, 2017; Gavory et al, 2018; Schauer et al, 2019). Some N-cyano pyrrolidines, which resemble known cathepsin C covalent inhibitors, have been reported in the patent literature to be dual inhibitors of UCHL1 and USP30 (Laine et al, 2011). High-throughput screening has also identified a racemic phenylalanine derivative as a USP30 inhibitor (Kluge et al, 2018). However, the specificity and biological activity of this compound has so far been only characterised superficially.

Here, we introduce FT3967385 (hereafter FT385), a modified N-cyano pyrrolidine tool compound USP30 inhibitor. We carefully correlate its effects upon the proteome and ubiquitylome of neuroblastoma SH5YSY cells, expressing endogenous Parkin. We also show that this compound can recapitulate effects of USP30 deletion on mitophagy and regulate the ubiquitylation status of translocase of the outer mitochondrial membrane (TOM) complex components. The TOM complex functions as a common entry portal for mitochondrial precursor proteins (Wiedemann & Pfanner, 2017). We propose that associated ubiquitin may provide nucleating sites at which PINK1 phosphorylation sets in train a feed-forward loop of further Parkin-mediated ubiquitylation (Marcassa et al, 2018). Accordingly, pUb generation after mitochondrial depolarisation is enhanced by both USP30 deletion and by inhibitor treatment.

# Results

We developed a tool compound inhibitor (FT385) for investigation of USP30 biology (Fig 1A). It shows a calculated IC$_{50}$ of ~1 nM in vitro using purified USP30, together with ubiquitin–rhodamine as a fluorogenic substrate (Fig 1B and E). Bio-layer interferometry experiments show binding behaviour that is consistent with covalent

modification of USP30 (Fig 1C) as indicated by other studies of cyano pyrrolidine inhibitors of USPs (Bashore et al, 2020). Progress curves for ubiquitin–rhodamine processing by USP30 were used to determine $K_I$ and $k_{inact}$ (Fig 1D and E). To test for selectivity of the inhibitor within the USP family of enzymes, we used the Ubiquigent DUB profiler screen, which tests inhibitory activity against a broad panel of USP enzymes. At the indicated concentrations (up to 200 nM), the inhibitor was highly selective for USP30 (Fig 1F). Only one other family member, the plasma membrane–associated USP6, showed a significant degree of inhibition (Urbe et al, 2012). This particular deubiquitylase shows a highly restricted expression profile (Barretina et al, 2012). It is not found in any of our deep proteome data sets nor was it identified in two recent studies that used state-of-the-art enrichment with active site probes to generate an inventory of cellular DUBs (Hewings et al, 2018; Pinto-Fernandez et al, 2019).

We used the competition between FT385 and Ub-propargylamide (Ub-PA), which covalently binds to the USP30 active site, to assess target engagement (Ekkebus et al, 2014). Binding of the probe to a DUB leads to an up-shift in apparent molecular weight on SDS–PAGE gels (Fig 2). If a drug is present that occupies or otherwise occludes this site, probe modification is inhibited and the protein mass is down-shifted accordingly. Our results demonstrate target engagement and allow us to determine a suitable concentration range for further experiments (Fig 2). In SHSY5Y neuroblastoma cells, effective competition of drug towards added probe is seen at concentrations >100 nM when added to cell lysates (Fig 2A) or pre-incubated with cells prior to lysis (Fig 2B).

To be able to compare compound activity to USP30 loss, we used CRISPR/Cas9 to generate YFP-Parkin-RPE1 (retinal pigment epithelium) and SHSY5Y (neuroblastoma) USP30 KO cells (Fig S1). We have previously shown that USP30 physically interacts with TOM20, a component of the OMM transport complex that recognises mitochondrial targeting sequences (Liang et al, 2015; Wiedemann & Pfanner, 2017). USP30 represses both depolarisation-induced mitophagy and the specific ubiquitylation of TOM20 in cells overexpressing Parkin (Bingol et al, 2014; Cunningham et al, 2015; Liang et al, 2015; Gersch et al, 2017). Application of FT385 to RPE1 cells overexpressing YFP–Parkin results in enhanced ubiquitylation and apparent loss of TOM20 without affecting PINK1 protein levels (Fig 3A). Enhancement of TOM20 ubiquitylation by FT385 under depolarising conditions is more clearly shown in Fig 3B. In this experiment, a shorter depolarisation time (1 h) has been used, at which there is minimal TOM20 loss to mitophagy or other pathways. USP30 KO and inhibitor-treated cells show similar elevation of ubiquitylated TOM20, whereas no further enhancement is achieved by inhibitor treatment of KO cells (Fig 3B). Thus, the TOM20 ubiquitylation response depends on USP30 catalytic activity and represents an on-target effect of the drug.

We confirmed that both USP30 deletion and inhibition can also lead to the accumulation of ubiquitylated TOM20 in SHSY5Y cells, both in whole cell lysates and in crude mitochondrial fractions (MFs) (Figs 3C and D and S2A–F). Here, we are detecting this modification without Parkin overexpression. TOM20 is atypical in the respect that we do not observe USP30-dependent changes to the ubiquitylation pattern of another mitochondrial Parkin substrate mitofusin 2 (MFN2) (Figs 3A, C, and D and S2A). To determine

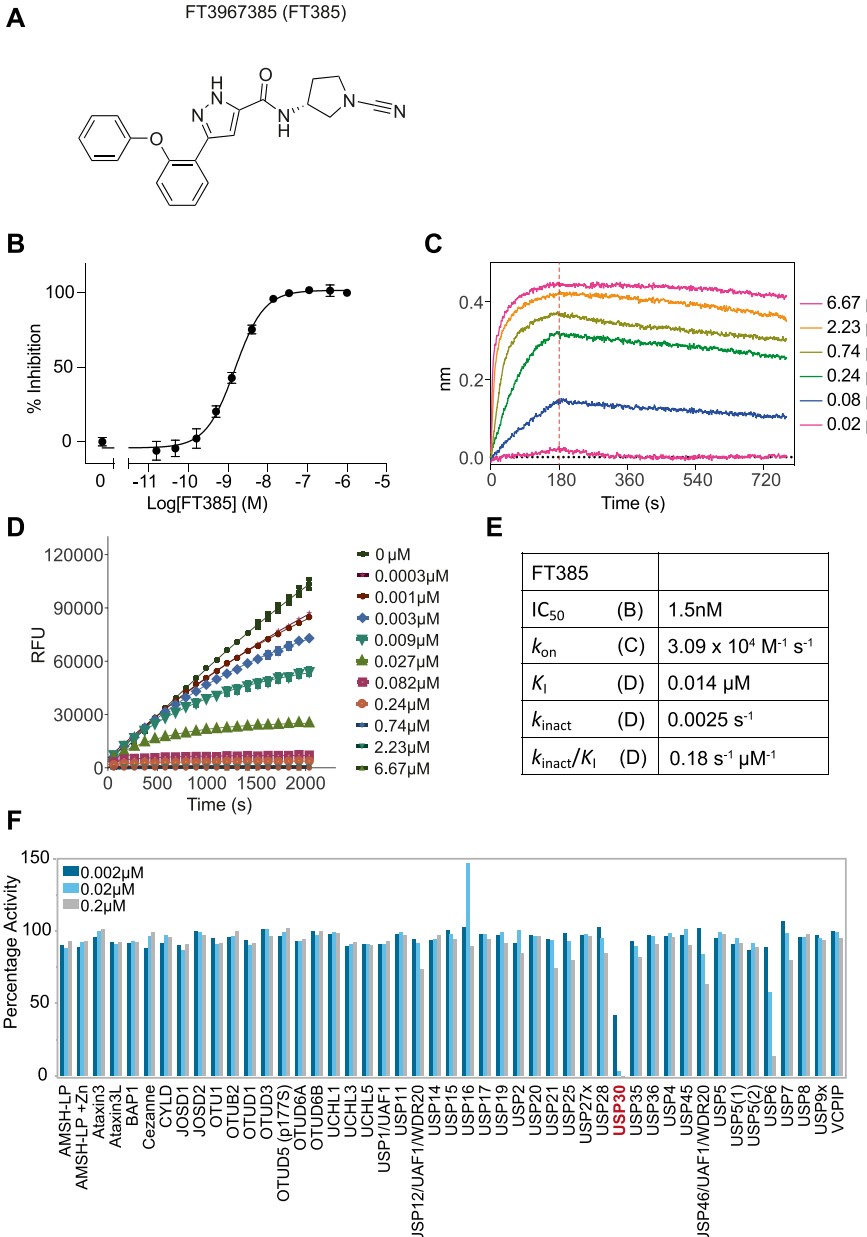

**Figure 1. FT3967385 is a selective covalent USP30 inhibitor.**
**(A)** Chemical structure of FT3967385 (FT385). **(B)** Concentration-dependent inhibition of recombinant USP30 activity using ubiquitin–rhodamine as a substrate. **(C)** Bio-layer interferometry traces showing no significant off-rate at indicated concentrations. Red line indicates removal of the inhibitor after 180 s. **(D)** Progress curves characteristic of a covalent inhibitor (0–6.67 μM), these are fitted to obtain $K_I$ and $k_{inact}$. **(E)** Data table of inhibitory properties. **(F)** DUB specificity screen (DUB profiler; Ubiquigent) with 2, 20, and 200 nM FT385.

| FT385 | | |
|---|---|---|
| IC$_{50}$ | (B) | 1.5nM |
| $k_{on}$ | (C) | 3.09 x 10$^4$ M$^{-1}$ s$^{-1}$ |
| $K_I$ | (D) | 0.014 μM |
| $k_{inact}$ | (D) | 0.0025 s$^{-1}$ |
| $k_{inact}/K_I$ | (D) | 0.18 s$^{-1}$ μM$^{-1}$ |

effects of USP30 inhibition on basal mitophagy, we used SHSY5Y cells expressing a tandem mCherry–GFP tag attached to the OMM localisation signal of the protein FIS1 (mitoQC) (Allen et al, 2013). A clear increase in the number of mitolysosomes per cell, indicative of increased mitophagic flux, is apparent after USP30 inhibition over a 96 h time period (Fig 3E).

Trypsin digestion of ubiquitylated proteins generates peptides with a residual diGly motif, which provides a characteristic mass shift and can be used for enrichment by immunoprecipitation (Peng et al, 2003). Several studies have used this approach to define Parkin substrates through proteomic analysis, after mitochondrial depolarisation in cell lines overexpressing Parkin (Sarraf et al, 2013; Ordureau et al, 2014, 2018). To search for potential substrates and/ or biomarkers beyond TOM20, we decided to take an unbiased view

of USP30 control of the cellular proteome and ubiquitylome in SHSY5Y cells, which endogenously express Parkin. Our experimental design, using triplexed combinations of SILAC labels, allowed quantitative comparison of both USP30 inhibitor treated (200 nM) and USP30 KO relative to parental untreated cells in basal conditions (proteome) or following mitochondrial depolarisation (proteome + ubiquitylome) (Fig 4A). We quantitated 6,423 proteins and 9,536 diGly peptides (which indicate specific sites of ubiquitylation), derived from 2,915 proteins (Table S1). We had hoped that the proteome might provide a biomarker that could be used in preclinical models for testing drug efficacy. Despite obtaining deep proteome coverage, we identified few proteins that responded to both genetic deletion and inhibition of USP30 (24 h) in a consistent manner across experiments. No impact of USP30 on total

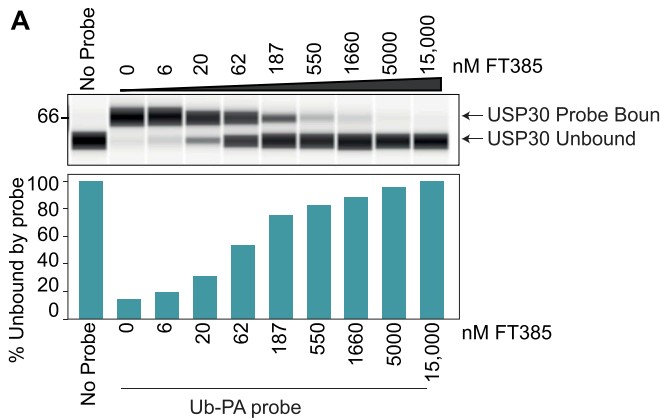

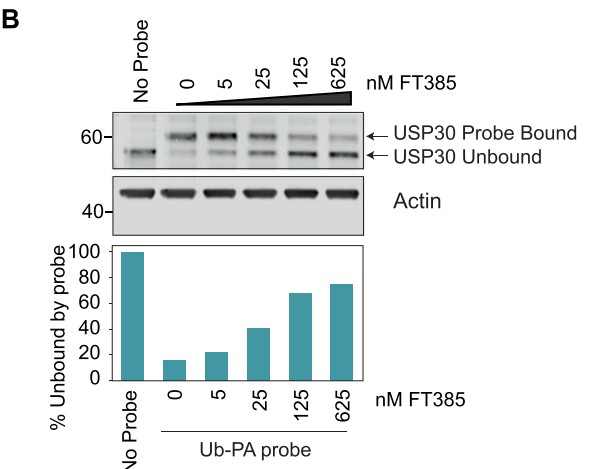

**Figure 2. Activity-based ubiquitin probe assay shows that FT385 engages USP30 in cells at low nanomolar concentrations.**
**(A, B)** SHSY5Y cell homogenates or (B) intact SHSY5Y cells were incubated with FT385 for 30 min or 4 h, respectively, at the indicated concentrations, and then incubated with Ub-PA probe for 15 min at 37°C and immunoblotted as shown. Samples in (A) were analysed using an automated Western blot (WES) system. Source data are available for this figure.

mitochondrial or peroxisomal mass after 24 h depolarisation is apparent (Fig 4B). This is in keeping with our observations and previous findings that in cell lines expressing endogenous levels of Parkin, the extent of depolarisation-induced mitophagy is low (Rakovic et al, 2013). In this experiment, we find that USP30 influences the ubiquitylation status of a small minority of proteins after depolarisation (Fig 4C). Most prominent among them are members of the voltage-dependent anion channel (VDAC) family. VDAC1, VDAC2, and VDAC3 show enhanced ubiquitylation at specific sites in the absence of USP30 activity without any change at the proteome level. In general, the effect is stronger in the USP30 KO cells but the pattern is conserved with USP30 inhibitor treatment (Figs 4C–E and S3A and B). Some proteins show a response to inhibitor but not to genetic loss of USP30 (for details see Table S1 and Fig S3C). There is no obvious connection between these proteins or enrichment for mitochondrial annotation, and they likely represent off-target effects. One conclusion from these data is that the global impact of USP30 activity at both the proteome and ubiquitylome levels is subtle. This makes pharmacology in both terminally differentiated

cellular models (e.g., primary cultured rodent neurons or human induced pluripotent stem cell-derived neurons) and in vivo experiments challenging. However, it is consistent with low impact on cell viability seen in CRISPR screens (Hart et al, 2017) and may in fact be a desirable feature of a drug target for a neurodegenerative disease.

To obtain information on the early USP30-dependent changes to the mitochondrial ubiquitylation profile that follow depolarisation, we compared two USP30 KO SHSY5Y clones with wild-type cells, using a shorter depolarisation period (4 h, Fig 5A). No systematic changes in mitochondrial or peroxisomal protein abundance were observed (Fig 5B). For the ubiquitylome arm of this experiment, we used crude MFs to increase coverage of specific mitochondrial components. This is evident in Fig 5C and D, which summarise the major changes in ubiquitylation we have identified at specific sites in both sets of experiments (Figs 4A, 5A, S3D and E, and Tables S1 and S2). Multiple responsive VDAC peptides were once again identified. Strong outliers are found in ganglioside-induced differentiation associated protein 1 (GDAP1), an OMM protein, mutations of which are linked to Charcot–Marie–Tooth neuropathy and mitochondrial dysfunction (Barneo-Munoz et al, 2015) and the mitochondrial outer membrane protein synaptojanin 2-binding protein (SYNJ2BP, Fig 5D and E) (Nemoto & De Camilli, 1999). Also prominent is peptidyl-tRNA hydrolase 2 (PTRH2), a mitochondrial protein linked to the release of non-ubiquitylated nascent chains from stalled ribosomal complexes (Kuroha et al, 2018). The improved coverage now also reveals USP30-dependent ubiquitylation of multiple TOM complex components, including the two translocase receptors, TOM20 and TOM70, the TOM40 channel and an accessory subunit TOM5 within this set of strong outliers.

In healthy mitochondria, PINK1 is imported through the TOM complex and subsequently cleaved and released for proteasomal degradation in the cytosol. In depolarised mitochondria, it is no longer imported and degraded but remains associated with TOM complex components on the OMM (Lazarou et al, 2012; Okatsu et al, 2013, 2015; Sekine & Youle, 2018). At this point, it becomes transactivated and initiates a signalling cascade by phosphorylating ubiquitin on Ser65 (generating pUb). This accumulation of pUb can be readily visualised by Western blotting using a specific antibody. We find that genetic loss of USP30 or USP30 inhibition both lead to a more rapid accumulation of pUb after mitochondrial depolarisation, without an evident increase in total PINK1 nor Parkin levels at mitochondria (Figs 6A–D and S4).

## Discussion

Here, we provide a comprehensive analysis of the impact of USP30 on mitochondrial ubiquitylation dynamics after mitochondrial membrane depolarisation. Our principal analysis is conducted on cells expressing endogenous levels of Parkin and we directly compare the effects of genetic loss with a specific inhibitor. This allows us to clearly attribute molecular signatures to catalytic activity for the first time. We have extended USP30 linkage to the mitochondrial import (TOM) complex to now include subunits beyond TOM20, which has been previously characterised (Liang et al, 2015; Bingol & Sheng, 2016; Gersch et al, 2017). We also identify

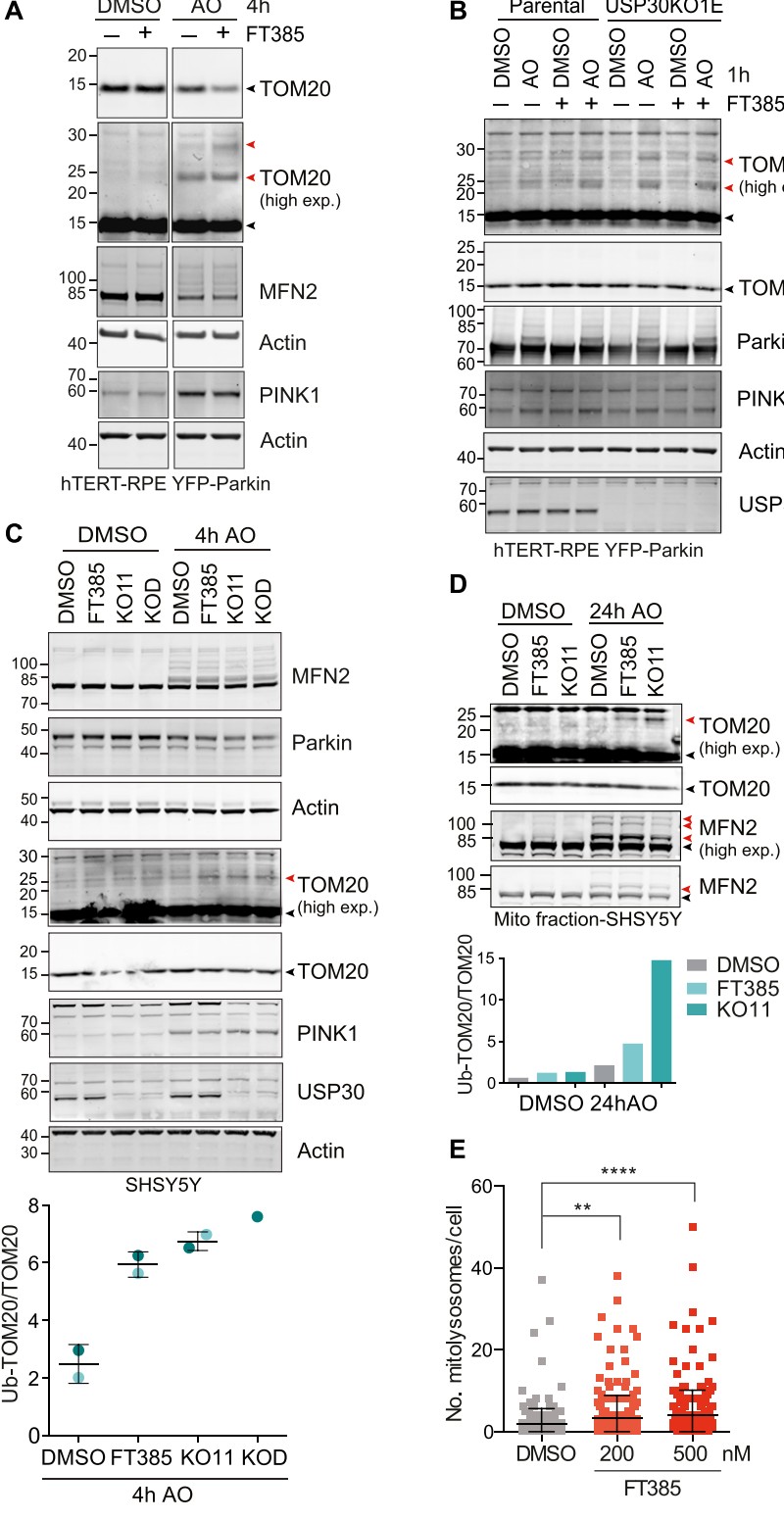

Figure 3.    Pharmacological inhibition of USP30 phenocopies USP30 KO in enhancing basal mitophagy and promoting ubiquitylation of TOM20 upon depolarisation.
**(A)** Inhibition of USP30 enhances the ubiquitylation and degradation of TOM20 in YFP–Parkin overexpressing hTERT-RPE1 cells in response to mitophagy induction. Cells were treated for 4 h with DMSO or antimycin A and oligomycin A (AO; 1 μM each) in the absence or presence of 200 nM FT385, lysed, and analysed by Western blotting. **(B)** USP30 inhibitor (FT385) treatment of parental YFP-Parkin overexpressing hTERT-RPE1 cells phenocopies USP30 deletion (KO1E) by promoting TOM20 ubiquitylation. In contrast, TOM20 ubiquitylation is unaffected by FT385 in the USP30 KO (KO1E) cells. Cells were treated for 1 h with or without AO (1 μM) in the absence or presence of 200 nM FT385, lysed, and samples analysed by immunoblotting. **(C)** TOM20 ubiquitylation is enhanced by USP30 inhibition and deletion in SHSY5Y cells expressing endogenous Parkin. SHSY5Y with or without FT385 (200 nM) and USP30 CRISPR/Cas9 KO cells (KO11 and KOD, two distinct sgRNAs) were treated with AO (1 μM each) for 4 h as indicated. Cells were then lysed and samples analysed by immunoblotting as shown. Graph shows quantification of ubiquitylated TOM20 normalised to unmodified TOM20 for two independent experiments with individual data points shown in dark and light blue. Error bars indicate the range. **(D)** SHSY5Y (mitoQC) and USP30 KO cells (KO11) were treated for 24 h with AO (1 μM each) in the presence or absence of FT385 (100 nM). Cells were subjected to subcellular fractionation and the mitochondrial fraction (MF) was analysed by immunoblotting as indicated. Bar chart shows quantification of ubiquitylated TOM20 normalised to unmodified TOM20. **(A, B, C, D)** Black and red arrowheads indicate unmodified and ubiquitylated TOM20 or MFN2 species, respectively (high exp, higher exposure). **(E)** Quantification of the number of mitolysosomes in SHSY5Y-mitoQC cells, treated with DMSO or FT385 (200 or 500 nM) for 96 h before imaging. Average ± SD; n = 3 independent experiments; 80 cells per experiment; one-way ANOVA with Dunnett's multiple comparisons test, **P < 0.01, ****P < 0.0001.
Source data are available for this figure.

a further substrate, SYNJ2BP, whose enhanced ubiquitylation can be monitored by Western blotting. Based on our studies, FT385 emerges as a promising tool compound for the study of USP30 biology. When used at appropriate concentrations, a high degree of specificity amongst DUB family members can be achieved. On the other hand, there are some inevitable liabilities; after inhibitor treatment, we identified several proteins with enhanced ubiquitylation that are not evident with genetic loss of USP30.

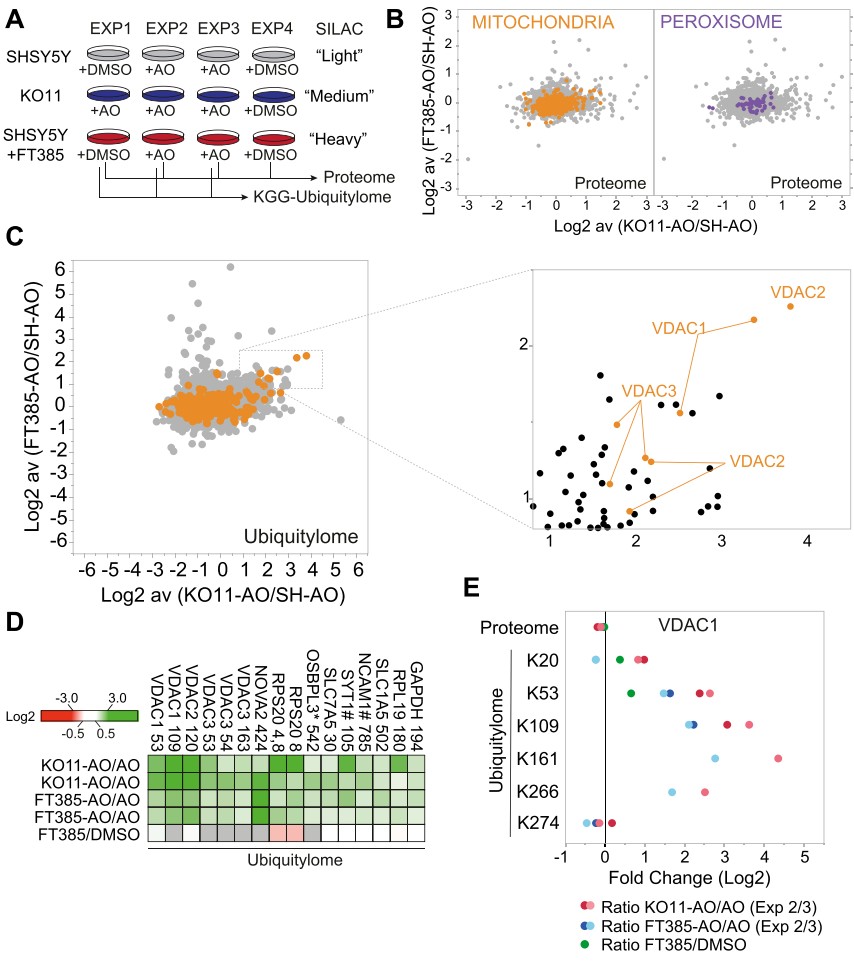

**Figure 4. Comparison of proteome and ubiquitylome changes in USP30 KO versus USP30 inhibitor treated SHSY5Y cells.**

**(A)** Schematic flow chart of SILAC based quantitative ubiquitylome and proteome analysis comparing USP30 KO and USP30 inhibition. SHSY5Y (USP30 wild-type) and SHSHSY USP30 KO (KO11) cells were metabolically labelled by SILAC as shown. Cells were then treated for 24 h with DMSO or antimycin A and oligomycin A (AO; 1 μM each) and/or FT385 (200 nM) as indicated. Cells were lysed and processed for mass spectrometry analysis. **(B, C)** Graphs depicting the fold change (log₂) in the proteome (B) or ubiquitylome (C) of AO-treated SHSY5Y cells ± FT385 treatment (y-axis) and ± USP30 (x-axis). Mitochondrial (Integrated Mitochondrial Protein Index database; http://www.mrc-mbu.cam.ac.uk/impi; "known mitochondrial" only) and peroxisomal proteins (peroxisomeDB; http://www.peroxisomedb.org) proteins are highlighted in orange and purple, respectively. Inset in (C) shows enlarged section of ubiquitylome data for peptides enriched in USP30 KO and inhibitor treated cells. **(B)** Within proteome graphs (B) each dot represents a single protein identified by at least two peptides and the ratio shows the average of two experiments. **(C)** Within ubiquitylome graphs (C) each dot represents a single diGly peptide (localisation ≥ 0.75) and the ratio shows the average of two experiments. **(D)** Heat map showing diGly peptides that are increased consistently by log₂ ≥ 0.8 in both USP30 KO and USP30 inhibitor (FT385) treated cells. Grey indicates the protein was not seen in that condition, * indicates ambiguity of peptide assignment between family members (OSBPL3, OSBPL7, and OSBPL6), # indicates an increase at proteome level in KO11. VDAC3 K53 and K54 correspond to equivalent lysines in two distinct isoforms. **(E)** Fold change (log₂) in proteome and individual diGly peptides (localisation ≥ 0.75) by site in VDAC1 proteins. **(D)** See Fig S3 for corresponding data sets for VDAC2 and 3 and proteome data for hits shown in (D).

Previous studies have suggested that the overall pattern of depolarisation-induced ubiquitylation of mitochondria is largely unchanged following USP30 knock-down, with TOM20 being an exception (Liang et al, 2015; Gersch et al, 2017). We see enhanced pUb accumulation in the absence of USP30 activity, despite the published observations that pUb-modified chains provide a poor substrate for USP30 (Wauer et al, 2015b; Gersch et al, 2017). How then might USP30 suppress mitophagy, as previously reported in several studies (Bingol et al, 2014; Cunningham et al, 2015; Liang et al, 2015; Marcassa et al, 2018)? We have previously shown that USP30 depletion enhances PINK1-dependent basal mitophagy even in the absence of Parkin (Marcassa et al, 2018). We and others have proposed that USP30 may regulate the availability of ubiquitin on specific trigger proteins that are most readily available for phosphorylation by PINK1. In other words, USP30 may determine the probability that a local accumulation of PINK1 can trigger feed-forward mechanisms that lead to mitophagy (Clague & Urbe, 2017; Gersch et al, 2017; Marcassa et al, 2018). The prominence of TOM complex components within the limited set of USP30-responsive diGly-peptides, and the known interaction with both USP30 (Liang et al, 2015) and with PINK1 (Lazarou et al, 2012; Okatsu et al, 2013, 2015; Sekine & Youle, 2018) suggest that this may be a critical pUb nucleation site regulated by USP30 (Fig 7).

While our manuscript was in preparation, two complementary studies have been published that also highlight the centrality of the TOM complex to USP30 function (Ordureau et al, 2020; Phu et al, 2020). All three studies use global proteome and ubiquitylome profiling. Ordureau et al (2020) examine the impact of USP30 genetic loss in iNeurons ± AO. Phu et al (2020) focus on basal conditions (no depolarisation agents) and use HEK293 cells to compare genetic loss with a USP30 inhibitor that is related to the one we describe here. Note that, in that latter study, a much higher concentration of inhibitor has been used (5 μM versus 200 nM). We identify an overlapping set of USP30-sensitive ubiquitylation sites with these studies. Our findings are more directly comparable with Ordureau et al (2020), as our data derive from AO-treated cells. For the majority (15/16) of USP30-sensitive mitochondrial proteins we describe in our ubiquitylome analysis (Fig 5C), corresponding increases have also been found in USP30 KO iNeurons, albeit the specific sites differ in some instances (Ordureau et al, 2020). Both studies find greater prevalence, than we do here, of elevated ubiquitylation of mitochondrial matrix and inner mitochondrial membrane proteins, although we do see a few examples of the same phenomenon (e.g., MDH2, GRSF1, and MTLN). Although ubiquitylation can occur within mitochondria (Lavie et al, 2018), USP30 is an OMM protein whose catalytic activity is facing towards

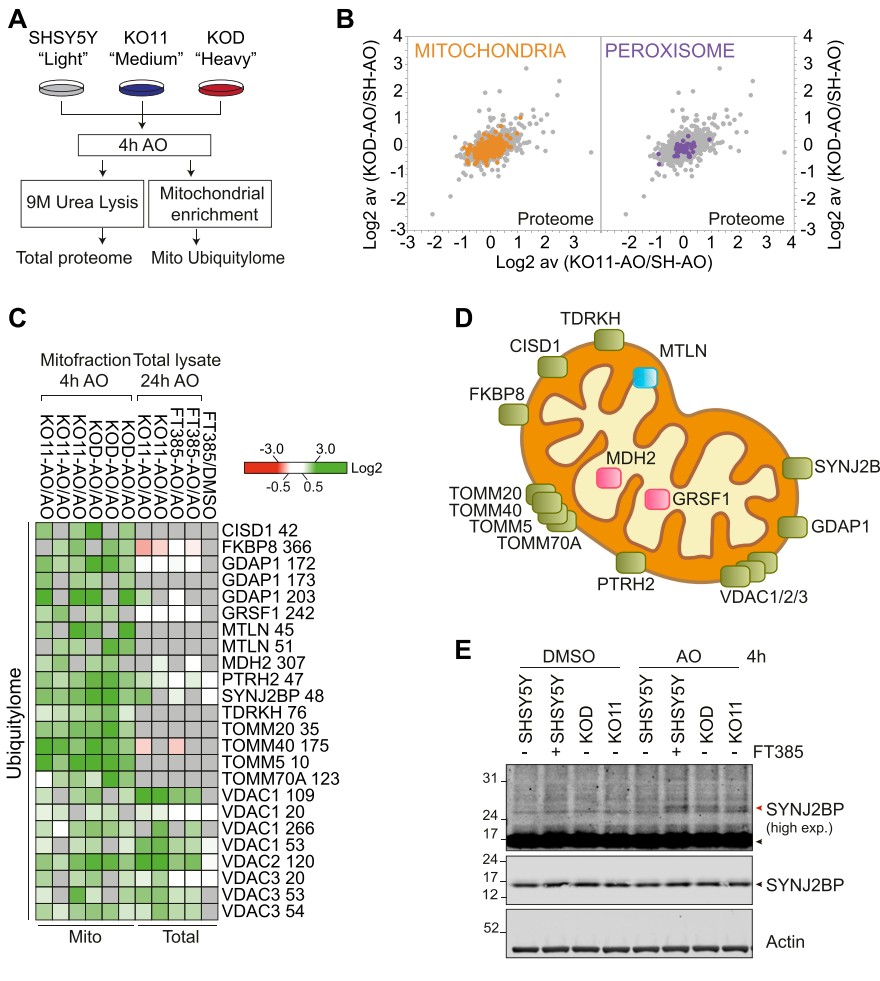

**Figure 5. Proteomic analysis of the mitochondria-enriched ubiquitylome in USP30 KO SHSY5Y cells.**
**(A)** Schematic flow chart of SILAC based quantitative ubiquitylome and proteome analysis comparing two USP30-KO clones (KOD-sgRNA#1 and KO11-sgRNA#2) to wild-type SHSY5Y cells. Cells were metabolically labelled by SILAC as shown and treated for 4 h with AO (1 $\mu$M). Cells were then either lysed for total proteome analysis or further processed by subcellular fractionation. The mitochondrial fraction was used as the starting material for the ubiquitylome analysis. **(B)** Graphs depicting the fold change (log$_2$) in the proteome of AO-treated USP30 KOD versus wild-type SHSY5Y (SH) (y-axis) and USP30 KO11 compared with SHSY5Y cells (x-axis). Mitochondrial (Integrated Mitochondrial Protein Index database; http://www.mrc-mbu.cam.ac.uk/impi; "known mitochondrial" only) and peroxisomal proteins (peroxisomeDB; http://www.peroxisomedb.org) proteins are highlighted in orange and purple, respectively. Each dot represents a single protein identified by at least two peptides and the ratio shows the average of three experiments. **(C)** Heat map showing diGly containing peptides that are increased consistently in at least four of six experimental conditions by log$_2 \geq 0.8$. The corresponding data from the total ubiquitylome experiment shown in Fig 4 are also indicated. Grey indicates the protein was not seen in that condition. VDAC3 K53 and K54 correspond to equivalent lysines in two distinct isoforms. **(D)** Depiction of the localisation of USP30 sensitive depolarisation-induced ubiquitylated proteins within mitochondria (enriched proteins shown in (C)). Defined as outer mitochondrial membrane (green), inner mitochondrial membrane (blue), or matrix (pink). **(E)** Western blot showing the appearance of mono-ubiquitylated species of SYNJ2BP in both USP30 KO clones (KO11 and KOD) and in USP30 inhibitor (FT385) treated cells. Cells were treated for 4 h with AO (1 $\mu$M) in the presence or absence of 200 nM FT385, then lysed in urea lysis buffer and analysed by Western blot. Black and red arrowheads indicate unmodified and ubiquitylated SYN2BP (high exp, higher exposure).
Source data are available for this figure.

the cytosol (Nakamura & Hirose, 2008; Marcassa et al, 2018). Hence, it has been suggested that this reflects ubiquitylation of newly synthesised proteins engaging with the TOM complex (Ordureau et al, 2020; Phu et al, 2020). Thus, USP30 might sit at the gate of the import complex pore and strip off ubiquitin as a prerequisite for entry. This provides a striking parallel with the action of proteasomal deubiquitylases, which control entry to the proteasome core (Lee et al, 2011). Ribosomes themselves interact directly with the TOM complex (Gold et al, 2017), and ribosomal quality control mechanisms have extensive links to the ubiquitin system (Joazeiro, 2017). Perturbation of these pathways, could also lead to a higher representation of ubiquitylated peptides derived from nascent imported proteins. Our finding that the mitochondrial peptidyl-tRNA hydrolase PTRH2 is a USP30 substrate provides a first link to ribosomal quality control. PTRH2 can cleave nascent chain tRNA on stalled ribosomes and provide a release mechanism for non-ubiquitylated nascent chains (Kuroha et al, 2018).

The USP30-dependent suppression of mitophagy is well established for events which rely on the overexpression of Parkin, together with acute mitochondrial depolarisation (Bingol et al, 2014; Cunningham et al, 2015; Liang et al, 2015). In fact, in a recently published whole genome screen for mitophagy regulators in Parkin overexpressing C2C12 myoblasts, USP30 is the most prominent mitochondrial annotated negative regulator (Hoshino et al, 2019). Our study contributes to a body of evidence that translates these findings to systems with endogenous Parkin expression levels (Marcassa et al, 2018; Ordureau et al, 2020; Phu et al, 2020). The physiological defects associated with PINK1/Parkin loss of function in PD are likely to accumulate slowly. The benign effects of USP30 loss or inhibition make it a target candidate that can be considered for long-term therapy. The availability of specific tool compounds, such as described here, will enable preclinical assessment of this strategy.

# Materials and Methods

## Cell culture

hTERT-RPE1-YFP-PARKIN (Liang et al, 2015), SHSY5Y, and SHSY5Y-mitoQC (mCherry-GFP-Fis1(101-152)) (Allen et al, 2013) cells were routinely cultured in Dulbecco's Modified Eagle's medium DMEM/F12 supplemented with 10% FBS and 1% non-essential amino acids.

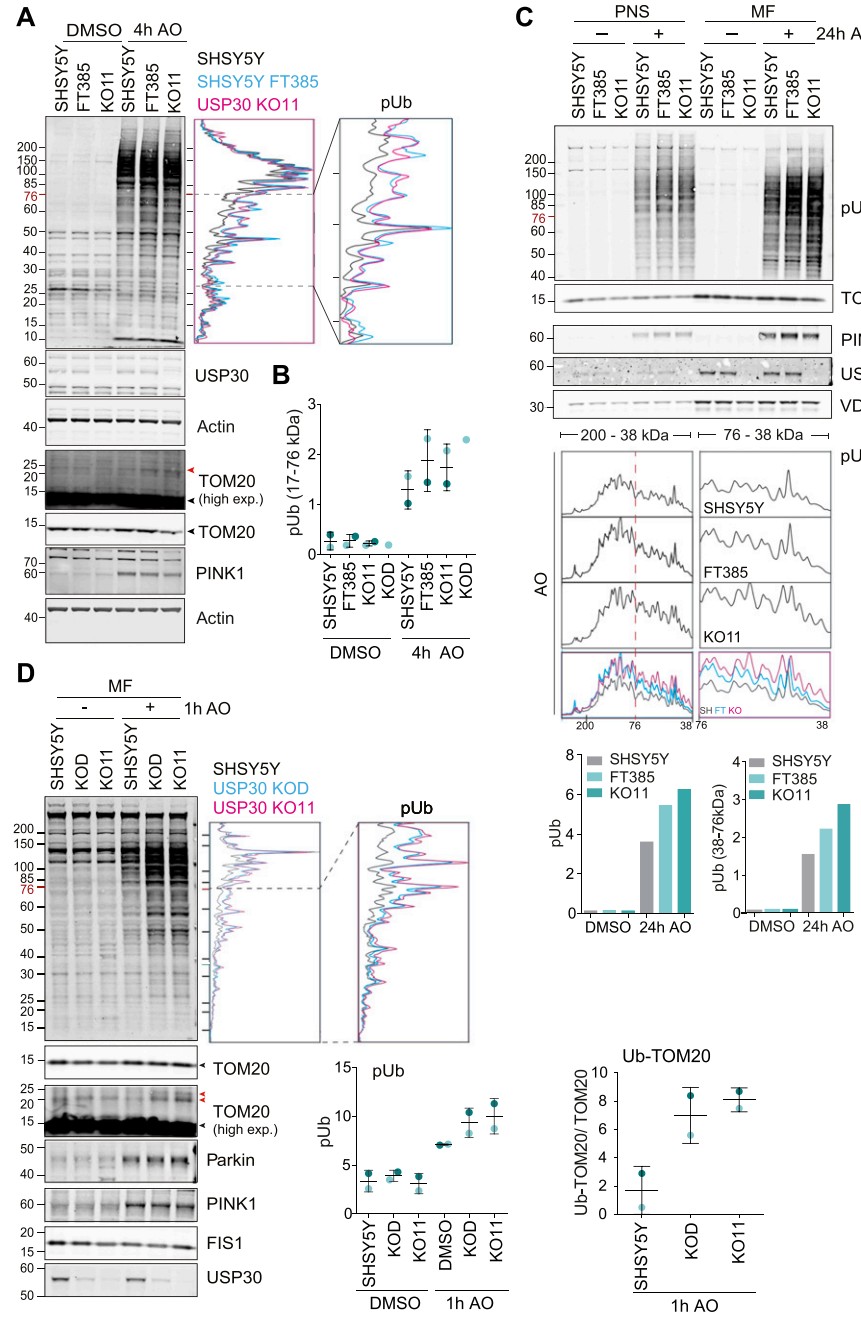

**Figure 6. USP30 KO and USP30 Inhibition enhance phospho-Ser65 ubiquitin levels on mitochondria of SHSY5Y cells.**

**(A)** Comparison of depolarisation induced phospho-Ser65 ubiquitin (pUb) generation in SHSY5Y cells treated with FT385 and in USP30 KO SHSY5Y (KO11). Shown is a Western blot and corresponding line graph for the pUb signal, of lysates from cells treated for 4 h with AO (1 $\mu$M) with or without FT385 (200 nM). Black and red arrowheads indicate unmodified and ubiquitylated TOM20 species, respectively (high exp, higher exposure). **(B)** Graph shows quantification of the pUb signal in the 17–76 kD range for two independent experiments (A, and Fig S4A) with individual data points shown in dark and light blue. Error bars indicate the range. **(C)** A post-nuclear supernatant and mitochondrial fractions were obtained from SHSY5Y cells treated in the presence or absence of FT385 (100 nM, 24 h), with DMSO or AO (1 $\mu$M). Samples were analysed by Western blotting and a line graph depicting the pUb signal is shown. Bar chart shows quantification of the total pUb signal (left) and the pUb signal in the 38–76 kD range (right). **(D)** SHSY5Y cells and two USP30 KO clones (KOD and KO11) were treated for 1 h with AO (1 $\mu$M). Cells were homogenised and mitochondrial fractions prepared and analysed as indicated. Graphs show quantification of the total pUb signal and the ubiquitylated TOM20 (red arrowheads) normalised to unmodified TOM20 (black arrowheads) for two independent experiments with individual data points shown in dark and light blue. Error bars indicate the range. High exp, higher exposure.

Source data are available for this figure.

## Generation of USP30 KO cells

USP30 KO cells were generated using CRISPR-Cas9 with USP30-specific sgRNAs targeting exon 3 of isoform 1 (sgRNA1: AGTT-CACCTCCCAGTACTCC, sgRNA2: GTCTGCCTGTCCTGCTTTCA). sgRNAs were cloned into the pSpCas9(BB)-2A-GFP (PX458) vector (plasmid #48138 46; Addgene) or PX330-Puro (kind gift from Prof Ciaran Morrison, NUI Galway). hTERT-RPE1-YFP-Parkin USP30 KO Clone 1E and SHSY5Y clones KOC and KOD were engineered by transfecting the parental lines with pSpCas9(BB)-2A-GFP-sgRNA1, followed by FACS 24 h later (selection for GFP positive cells) and single cell dilution. SHSY5Y-mitoQC Clone 11 was engineered by transfection with PX330-Puro-sgRNA2 followed by selection with 1–1.5 $\mu$g/ml puromycin and single cell dilution. The positive clone (KO11) has lost expression of the mitoQC fluorophore. Individual clones of SHSY5Y KO cells were amplified and multiple alleles sequenced (Fig S1).

## Antibodies and reagents

Antibodies and other reagents used were as follows: anti-USP30 (HPA016952, 1:500; Sigma-Aldrich), anti-USP30 (PA5-53523, 1:1,000;

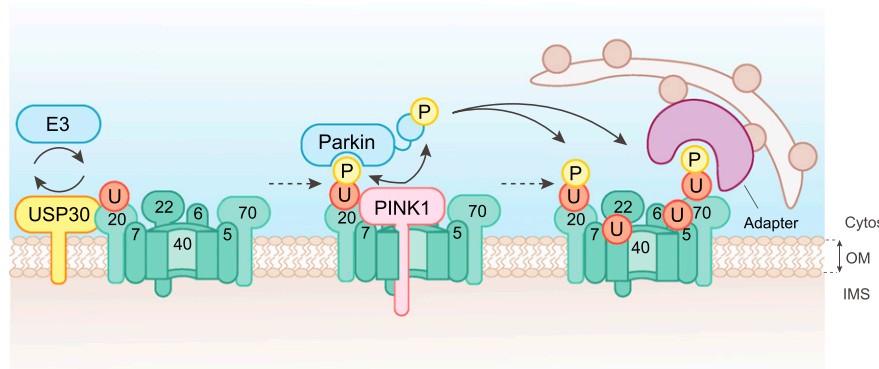

**Figure 7. Working model depicting USP30 action upstream of PINK1.**
Under depolarising conditions, PINK1 becomes activated but remains associated with TOM complex components. TOM complex associated ubiquitylation provides the nucleating substrate for PINK1-dependent phosphorylation of ubiquitin on Ser65. This leads to recruitment and activation of the E3 ligase Parkin, which can then amplify the signal. By opposing TOM complex ubiquitylation, USP30 suppresses the trigger for mitophagy.

Thermo Fisher Scientific), anti-USP30 (MRC PPU, 1:1,000), anti-USP30 (sc-515235, 1:1,000; Santa-Cruz), anti-PINK1 (D8G3, 6946S, 1:1,000; Cell Signalling Technology), anti-TOM20 (HPA011562, 1:1,000; Sigma-Aldrich), anti-PARK2 (sc32282, 1:250; Santa-Cruz), anti-MFN2 (ab56889, 1:1,000; Abcam), anti-ubiquitin (VU101, 1:2,000; Lifesensor), anti-FIS1 (10956-1-AP, 1:1,000; ProteinTech), anti-phospho-ubiquitin Ser65 (ABS1513-I, 1:1,000; Millipore), anti-phosphoubiquitin Ser65 (62802, 1:1,000; Cell Signalling Technology), anti-VDAC1 (ab15895, 1:1,000; Abcam), mouse anti-actin (ab6276, 1:10,000; Abcam), mouse anti-actin (66009-1-Ig, 1:10,000; ProteinTech), rabbit-anti-actin (20536-1-AP, 1:10,000; ProteinTech), anti-SYNJ2BP (HPA000866, 1:1,000; Sigma-Aldrich), oligomycin A (75351; Sigma-Aldrich), and antimycin A (A8674; Sigma-Aldrich).

### Preparation of cell lysates and Western blot analysis

Cultured cells were either lysed with urea buffer (Fig 6E, 9 M urea and 20 mM Hepes–NaOH, pH 7.4) supplemented with 2-chloro-acetamide (CAA; Sigma-Aldrich) or NP-40 (0.5% NP-40, 25 mM Tris–HCl, pH 7.5, 100 mM NaCl, and 50 mM NaF) lysis buffer and routinely supplemented with mammalian protease inhibitor (MPI) cocktail (Sigma-Aldrich) and Phostop (Roche), with the exception of data presented in Fig 2. Proteins were resolved using SDS–PAGE (Invitrogen NuPage gel 4–12%), transferred to nitrocellulose membrane, blocked in 5% milk, 5% BSA or 0.1% fish skin gelatin in TBS supplemented with Tween-20, and probed with primary antibodies overnight. Visualisation and quantification of Western blots were performed using IRdye 800CW and 680LT coupled secondary antibodies and an Odyssey infrared scanner (LI-COR Biosciences).

### Subcellular fractionation

SHSY5Y cells were washed with ice-cold PBS and then collected by scraping and centrifugation at 1,000$g$ for 2 min. Cell pellets were washed with HIM buffer (200 mM mannitol, 70 mM sucrose, 1 mM EGTA, and 10 mM Hepes–NaOH, pH 7.4) and then resuspended in HIM buffer supplemented with MPI. Cells were mechanically disrupted by shearing through a syringe with a 27G needle, followed by passing three times through an 8.02-mm-diameter "cell cracker" homogeniser using an 8.01-mm-diameter ball bearing (Aubry &

Klein, 2006) or passage through a 27G needle (Fig 2A). The resulting homogenate was cleared from nuclei and unbroken cells by centrifugation at 600$g$ for 10 min to obtain a post-nuclear super-natant (PNS). The PNS was separated into the post-mitochondrial supernatant and crude MF by centrifugation at 7,000$g$ for 15 min. The MF pellet was resuspended in HIM buffer + MPI.

### Activity probe assay

Cells were mechanically homogenised in HIM buffer supplemented with 1 mM DTT (Fig 2A) or 1 mM Tris(2-carboxyethyl)phosphine (TCEP, Fig 2B) to obtain the PNS. Homogenates were incubated with Ub-propargyl (Ub-PA) probe at 1:100 (w/w) for 15 min at 37°C (Ekkebus et al, 2014). The reaction was stopped by the addition of sample buffer and heating at 95°C. To test drug engagement, either intact cells or cell homogenate (PNS, without addition of protease inhibitors) were treated with FT385. Intact cells were treated for 4 h at 37°C before homogenisation, and the homogenate was pre-incubated for 30 min at room temperature before probe incubation. Samples were either processed using a WES system and transformed to a virtual Western blot (Fig 2A, Protein Simple, Biotechne) or analysed by standard Western blot (Fig 2B).

### SILAC labelling

SHSY5Y and SHSY5Y-KO11 cells were grown for at least eight pas-sages in SILAC DMEM/F12 supplemented with 10% dialysed FBS, 200 mg/l L-proline, and either L-lysine (Lys0) together with L-arginine (Arg0), L-lysine-$^2$H$_4$ (Lys4) with L-arginine-U-$^{13}$C$_6$ (Arg6), or L-lysine-U-$^{13}$C$_6$-$^{15}$N$_2$ (Lys8) with L-arginine-U-$^{13}$C$_6$-$^{15}$N$_4$ (Arg10) at final con-centrations of 28 mg/l arginine and 146 mg/l lysine.

### Proteomics methods

For the experiments shown in Fig 4, SILAC labelled cells were lysed by sonication in 9 M urea, 20 mM Hepes, pH 8.0, 1 mM sodium orthovanadate, 2.5 mM sodium pyrophosphate, and 1 mM glycerol-3-phosphate. In experiments 1 and 2 the "medium" samples (Fig 4A) are derived from the same lysate. For total proteome and ubiquitylome, 700 μg and 20 mg, respectively, of each sample was combined at a 1:1:1 ratio. For the experiments shown in Fig 5, MFs (ubiquitylome)

were obtained by homogenisation in HIM buffer supplemented with MPI, 50 mM CAA, and Phostop from SILAC labelled cells. Cell pellets (proteome) or MFs were lysed by sonication in 9 M urea, 20 mM Hepes, pH 8.0, 1.15 mM sodium molybdate, 1 mM sodium orthovanadate, 4 mM sodium tartrate dihydrate, 5 mM glycerol-3-phosphate, and 1 mM sodium fluoride, and then reduced and alkylated with either 4.5 mM dithiothreitol/10 mM iodoacetamide (Fig 4) or 10 mM TCEP/10 mM CAA (Fig 5). Urea was then diluted fourfold by the addition of 20 mM Hepes, pH 8.0, buffer before trypsinisation overnight. The resultant tryptic peptides were acidified with trifluoroacetic acid and purified on a C18 Sep-Pak column before lyophilisation (Fig 4) or drying with a SpeedVac (Fig 5).

For ubiquitylome samples, modified peptides were enriched by immunoprecipitation using a diGly specific antibody in accordance with the manufacturer's instructions (PTMScan Ubiquitin Remnant Motif [K-GG] Kit #5562; Cell Signalling Technology). Eluted peptides were purified using C18 stage tips (Fig 4) or C18 Sep-Pak columns (Fig 5). Samples were then dried in a SpeedVac before resuspension and analysis by nano ultra-performance liquid chromatography tandem mass spectrometry (LC–MS/MS). Ubiquitylome (Fig 4) samples were analysed (total five technical replicates) on an Orbitrap Fusion Lumos (one replicate) and Orbitrap Q Exactive HF (four replicates). Ubiquitylome (Fig 5) samples were analysed on an Orbitrap Fusion Lumos.

For proteome samples, peptides were separated by fractionation. For Fig 4, samples were fractionated by off-line high-pH reverse-phase pre-fractionation as previously described (Davis et al, 2017), with the exception that eluted peptides were concatenated down to 10 fractions. Briefly, digested material was fractionated using the loading pump of a Dionex Ultimate 3000 HPLC with an automated fraction collector and an XBridge BEH C18 XP column (3 × 150 mm, 2.5 μm particle size, Waters no. 186006710) over a 100-min gradient using basic pH reverse-phase buffers (A: water, pH 10 with ammonium hydroxide; B: 90% acetonitrile, pH 10 with ammonium hydroxide). The gradient consisted of a 12-min wash with 1% B, then increasing to 35% B over 60 min, with a further increase to 95% B in 8 min, followed by a 10-min wash at 95% B and a 10-min re-equilibration at 1% B, all at a flow rate of 200 μl/min with fractions collected every 2 min throughout the run. 100 μl of the fractions was dried and resuspended in 20 μl of 2% acetonitrile/0.1% formic acid for analysis by LC–MS/MS. Fractions were loaded on the LC–MS/MS (Orbitrap Q Exactive HF) after concatenation of 50 fractions into 10, combining fractions in a 10-fraction interval (F1 + F11 + F21 + F31 + F41... to F10 + F20 + F30 + F40 + F50). For Fig 5, samples were fractionated by off-line reverse-phase pre-fractionation using a Dionex Ultimate 3000 Off-line LC system. Briefly, digested material was fractionated using the loading pump of a Dionex Ultimate 3000 HPLC with an automated fraction collector and with a Gemini C18 (3 μm particle size, 110A pore, 3 mm internal diameter, 250 mm length, #00G-4439-Y; Phenomenex) over a 39-min gradient using the following buffers: A: 20 mM ammonium formate, pH = 8; B: 100% ACN. The gradient consisted of a 1-min wash with 1% B, then increasing to 35.7% B over 28 min, followed by a 5-min wash at 90% B and a 5-min re-equilibration at 1% B, all at a flow rate of 250 μl/minute with fractions collected every 45 s from 2 to 38 min for a total of 48 fractions. Non-consecutive concatenation of every 13th fraction was used to obtain 12

pooled fractions (Pooled Fraction 1: Fraction 1 + 13 + 25 + 27, Pooled Fraction 2: Fraction 2 + 14 + 26 + 38...) that were analysed by LC-MS/MS (Orbitrap Q Exactive HF).

**Orbitrap Q Exactive HF LC–MS/MS parameters**

Peptide fractions were analysed by nano-UPLC-MS/MS using a Dionex Ultimate 3000 nano-UPLC with EASY spray column (75 μm × 500 mm, 2 μm particle size; Thermo Fisher Scientific) with a 60-min gradient (Fig 4), a 140 min gradient (Fig 5 Exp1), or a 120 min gradient (Fig 5 Exp2/3) of 2–35% acetonitrile, 0.1% formic acid in 5% DMSO at a flow rate of ~250 nl/minute (Fig 4), or 0–28% acetonitrile, 0.1% formic acid in 3% DMSO at a flow rate of ~300 nl/minute (Fig 5). Mass spectrometry (MS) data were acquired with an Orbitrap Q Exactive HF instrument in which survey scans were acquired at a resolution of 60,000 (Fig 4) or 120,000 (Fig 5) at 200 m/z, and the 20 most abundant precursors were selected for higher energy collisional dissociation (HCD) fragmentation with a normalised collision energy of 28% (Fig 4) or 25% (Fig 5 Exp1) or 30% (Fig 5 Exp2/3).

**Orbitrap Fusion Lumos LC–MS/MS parameters**

Ubiquitome samples were analysed by LC–MS/MS on a Dionex Ultimate 3000 connected to an Orbitrap Fusion Lumos. For experiments presented in Fig 4, peptides were separated using a 60-min linear gradient from 2 to 35% acetonitrile in 5% DMSO and 0.1% formic acid at a flow rate of 250 nl/minute on a 50-cm EASY spray column (75 μm × 500 mm, 2 μm particle size; Thermo Fisher Scientific). For experiments presented in Fig 5, peptides were separated using 140 (Fig 5 Exp1) or 240 (Fig 5 Exp2/3) minute linear gradients from 0 to 28% acetonitrile in 3% DMSO, 0.1% formic acid at a flow rate of 300 nl/minute on a 50-cm EASY spray column (75 μm × 500 mm, 2 μm particle size; Thermo Fisher Scientific). MS1 scans were acquired at a resolution of 120,000 between 400 and 1,500 m/z with an AGC target of $4 \times 10^5$. Selected precursors were fragmented using HCD at a normalised collision energy of 28% (Fig 4) or 30% (Fig 5 Exp1) or 32% (Fig 5 Exp2/3), an AGC target of $4 \times 10^3$ (Figs 4 and 5, Exp2/3) or $1 \times 10^4$ (Fig 5 Exp1), a maximum injection time of 35 ms (Fig 4) or 45 ms (Fig 5 Exp1) or 50 ms (Fig 5 Exp2/3), a maximum duty cycle of 1 s (Fig 4) or 3 s (Fig 5), and a dynamic exclusion window of 60 s (Fig 4) or 35 s (Fig 5). MS/MS spectra were acquired in the ion trap using the rapid scan mode.

**MS data analysis**

All raw MS files from the biological replicates of the SILAC-proteome experiments were processed with the MaxQuant software suite; version 1.6.7.0 using the Uniprot database (retrieved in July 2019) and the default settings (Tyanova et al, 2016). Cysteine carbamidomethylation was set as a fixed modification, whereas oxidation, phospho(STY), GlyGly (K), and acetyl N terminal were considered as variable modifications. Data were requantified. ProteinGroup text files (proteome) or GlyGly (K) site files were further processed using Excel (see Table S1) and Perseus (version 1.6.10.50). Graphs were plotted using JMP13. Heat maps were generated using Morpheus (Broad Institute).

### In vitro USP30 activity assay

Fluorescence intensity measurements were used to monitor the cleavage of a ubiquitin–rhodamine substrate. All activity assays were performed in black 384-well plates in 20 mM Tris–HCl, pH 8.0, 0.01% Triton-X, 1 mM L-glutathione, and 0.03% bovine gamma globulin with a final assay volume of 20 $\mu$l. Compound $IC_{50}$ values for DUB inhibition were determined as previously described (Turnbull et al, 2017). Briefly, an 11-point dilution series of compounds were dispensed into black 384-well plates using an Echo (Labcyte). USP30, 0.2 nM (#E-582 residues 57-517; Boston Biochem), was added and the plates pre-incubated for 30 min. 25 nM ubiquitin–rhodamine 110 (Ubiquigent) was added to initiate the reaction, and the fluorescence intensity was recorded for 30 min on a Pherastar FSX (BMG Labtech) with a 485-nm excitation/520-nm emission optic module. Initial rates were plotted against compound concentration to determine $IC_{50}$.

### $k_{inact}/K_I$ determination

A $k_{inact}/K_I$ assay was carried out using 0.2 nM USP30 and 180 nM ubiquitin–rhodamine 110 as described above with the omission of the 30-min pre-incubation step. Upon addition of the substrate, fluorescence intensity was monitored kinetically over 30 min. Analysis was performed in GraphPad Prism. Kinetic progress curves were fitted to equation $y = (v_i/k_{obs}) (1 - \exp(-k_{obs}x))$ to determine the $k_{obs}$ value. The $k_{obs}$ value was then plotted against the inhibitor concentration and fitted to the equation $y = k_{inact}/(1 + (K_I/x))$ to determine $k_{inact}$ and $K_I$ values.

### Bio-layer interferometry

Bio-layer interferometry was performed on an Octet RED384 system (ForteBio) at 25°C in a buffer containing 50 mM Hepes buffer (pH 7.5), 400 mM NaCl, 2 mM TCEP, 0.1% Tween, 5% glycerol, and 2% DMSO. Biotinylated USP30 (residues 64-502Δ179-216 & 288-305; Viva Biotech Ltd.) was loaded onto SuperStreptavidin (SSA) biosensors. Association of defined concentrations of FT385 (0–6.67 $\mu$M) was recorded over 180 s followed by dissociation in buffer over 600 s. Traces were normalised by double subtraction of baseline (no USP30, no compound) and reference sensors (no USP30, association and dissociation of compound) to correct for non-specific binding to the sensors. Traces were analysed using Octet Software (Version 11.2; ForteBio).

### Live-cell imaging and basal mitophagy quantification

SHSY5Y cells stably expressing mCherry-GFP-Fis1 (101-152) (SHSY5Y mitoQC) (Allen et al, 2013) were treated every 24 h over a 96-h time course with 200 and 500 nM of FT385. Cells were re-plated onto an IBIDI $\mu$-Dish (2 × 10$^5$) 2 d before live-cell imaging with a 3i Marianas spinning disk confocal microscope (63× oil objective, NA 1.4, Photometrics Evolve EMCCD camera, Slide Book 3i v3.0). Cells were randomly selected using the GFP signal and images acquired sequentially (488 nm laser, 525/30 emission; 561 nm laser, 617/73 emission). Analysis of mitophagy levels was performed using the "mito-QC Counter" implemented in FIJI v2.0 software (ImageJ; NIH) as previously described (Montava-Garriga et al, 2020), using the

following parameters: radius for smoothing images = 1.25, ratio threshold = 0.8, and red channel threshold = mean + 0.5 SD. Mitophagy analysis was performed for three independent experiments with 80 cells per condition. One-way ANOVAs with Dunnett's multiple comparisons were performed using GraphPad Prism 6. *P*-values are represented as **$P < 0.01$, ****$P < 0.0001$. Error bars denote SD.

## Data Availability

The MS data from this publication have been deposited to the ProteomeXchange Consortium via the PRIDE partner repository and assigned the identifier PXD019692 (Data in Fig 4) and PXD018640 (Data in Fig 5).

## Supplementary Information

## Acknowledgements

We thank Jon Lane and Ian Ganley for provision of cell lines. Funding for development of FT385 was provided by Forma Therapeutics. Additional support was provided by Celgene, Michael J. Fox Foundation, Alzheimer's Research UK (M Giurrandino, E Murphy, K England), Parkinson's UK (H-1502 J Jardine), Medical Research Council (MR/N00941X/1 E Marcassa, A Kallinos, K McCarron), Wellcome Trust (FG Barone, A Gajbhiye), and European Union (F Lamoliatte).

### Author Contributions

EV Rusilowicz-Jones: data curation, formal analysis, investigation, and writing—original draft, review, and editing.
J Jardine: data curation, formal analysis, investigation, and writing—original draft, review, and editing.
A Kallinos: data curation, formal analysis, and investigation.
A Pinto-Fernandez: formal analysis, investigation, and writing—review and editing.
F Guenther: formal analysis and investigation.
M Giurrandino: formal analysis, investigation, and writing—review and editing.
FG Barone: formal analysis, investigation, and methodology.
K McCarron: formal analysis, investigation, and methodology.
CJ Burke: formal analysis and investigation.
A Murad: formal analysis and investigation.
A Martinez: investigation.
E Marcassa: investigation.
M Gersch: resources.
AJ Buckmelter: resources.
KJ Kayser-Bricker: resources and writing—review and editing.
F Lamoliatte: investigation and methodology.
A Gajbhiye: investigation and methodology.
S Davis: investigation and methodology.
HC Scott: investigation and methodology.
E Murphy: investigation and methodology.
K England: investigation and methodology.

H Mortiboys: conceptualization, funding acquisition, and writing—review and editing.

D Komander: conceptualization, funding acquisition, and writing—review and editing.

M Trost: supervision, funding acquisition, methodology, and writing—review and editing.

BM Kessler: conceptualization, supervision, funding acquisition, methodology, and writing—review and editing.

S Ioannidis: conceptualization, funding acquisition, and writing—review and editing.

MK Ahlijanian: conceptualization, supervision, funding acquisition, and writing—review and editing.

S Urbe: conceptualization, supervision, funding acquisition, and writing—original draft, review, and editing.

MJ Clague: conceptualization, supervision, funding acquisition, and writing—original draft, review, and editing.

## Conflict of Interest Statement

The authors declare that they have no conflict of interest.

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
