## [Reviewer comments · Life Science Alliance]

USP30 sets a trigger threshold for PINK1-PARKIN amplification of mitochondrial ubiquitylation

Emma Rusilowicz-Jones, Jane Jardine, Andreas Kallinos, Adan Pinto-Fernandez, Franziska Guenther, Mariacarmela Giurrandino, Francesco Barone, Katy McCarron, Christopher Burke, Alejandro Murad, Aitor Martinez, Elena Marcassa, Malte Gersch, Alex Buckmelter, Katherine Kayser-Bricker, Frederic Lamoliatte, Akshada Gajbhiye, Simon Davis, Hannah Scott, Emma Murphy, Katherine England, Heather Mortiboys, David Komander, Matthias Trost, Benedikt Kessler, Stephanos Ioannidis, Michael Ahlijanian, Sylvie Urbé and Michael Clague

DOI: 10.26508/lsa/202000768

Corresponding author(s): Prof. Michael J. Clague (University of Liverpool) and Sylvie Urbé (University of Liverpool)

Review timeline:

Submission Date:	2020-05-07
Editorial Decision:	2020-05-26
Revision Received:	2020-06-24
Editorial Decision:	2020-06-25
Revision Received:	2020-06-25
Accepted:	2020-06-26

Transaction Report:

No Peer Review Process File is available with this article, as the authors have chosen not to make the review process public in this case.

Re: Life Science Alliance manuscript #LSA-2020-00768-T

Prof. Michael J. Clague
University of Liverpool
Physiology, Biomedical Sciences
Crown St.
Liverpool, Merseyside L69 3BX
United Kingdom

Dear Dr. Clague,

Thank you for submitting your manuscript entitled "A novel USP30 inhibitor recapitulates genetic loss of USP30 and sets the trigger for PINK1-PARKIN amplification of mitochondrial ubiquitylation." to Life Science Alliance. The manuscript was assessed by expert reviewers, whose comments are appended to this letter.

As you will see, while the reviewers appreciate your work, they also think that your claims need further strengthening. We would like to invite you to submit a revised version of your manuscript to us, addressing the individual criticisms raised. Importantly, the data on TOM20 ubiquitylation need to get strengthened as well as your conclusions regarding pUb levels. Please also include a better characterization of the inhibitor as well as a more detailed analysis of the Mass-Spec results. Finally, the discussion should get expanded and more comparisons to the recent, related work included.

Thank you for this interesting contribution to Life Science Alliance. We are looking forward to receiving your revised manuscript.

Sincerely,

Andrea Leibfried, PhD
Executive Editor
Life Science Alliance
Meyerhofstr. 1
69117 Heidelberg, Germany
t +49 6221 8891 414
e contact@life-science-alliance.org
www.life-science-alliance.org

B. MANUSCRIPT ORGANIZATION AND FORMATTING:

RE: Life Science Alliance Manuscript #LSA-2020-00768-TR

Prof. Michael J. Clague
University of Liverpool
Physiology, Biomedical Sciences
Crown St.
Liverpool, Merseyside L69 3BX
United Kingdom

Dear Dr. Clague,

Thank you for submitting your revised manuscript entitled "USP30 sets the trigger for PINK1-PARKIN amplification of mitochondrial ubiquitylation.". We would be happy to publish your paper in Life Science Alliance pending final revisions necessary to meet our formatting guidelines.

- please have secondary corresponding author add their ORCID ID
- please add a conflict of interest statement to your manuscript
- For Figure S2, please add a callout to the Figure or all of the figure panels (you currently have a callout for Figure S2 & Figure S2A)
- please provide your tables in editable docx or excel format versions
- please list 10 authors and et al. in your references

A. FINAL FILES:

B. MANUSCRIPT ORGANIZATION AND FORMATTING:

Sincerely,

Reilly Lorenz
Editorial Office Life Science Alliance
Meyerhofstr. 1
69117 Heidelberg, Germany
t +49 6221 8891 414
e contact@life-science-alliance.org
www.life-science-alliance.org

RE: Life Science Alliance Manuscript #LSA-2020-00768-TRR

Prof. Michael J. Clague
University of Liverpool
Cellular and Molecular Physiology, Biomedical Sciences
Crown St.
Liverpool, Merseyside L69 3BX
United Kingdom

Dear Dr. Clague,

Thank you for submitting your Research Article entitled "USP30 sets a trigger threshold for PINK1-PARKIN amplification of mitochondrial ubiquitylation.". It is a pleasure to let you know that your manuscript is now accepted for publication in Life Science Alliance. Congratulations on this interesting work.

DISTRIBUTION OF MATERIALS:

Again, congratulations on a very nice paper. I hope you found the review process to be constructive and are pleased with how the manuscript was handled editorially. We look forward to future exciting submissions from your lab.

Sincerely,

Reilly Lorenz
Editorial Office Life Science Alliance

Meyerhofstr. 1
69117 Heidelberg, Germany
t +49 6221 8891 414
e contact@life-science-alliance.org
www.life-science-alliance.org